# Examining Saudi Physicians’ Approaches to Communicate Bad News and Bridging Generational Gaps

**DOI:** 10.3390/healthcare11182528

**Published:** 2023-09-13

**Authors:** Ahmed Saad Al Zomia, Hayfa A. AlHefdhi, Abdulrhman Mohammed Alqarni, Abdullah K. Aljohani, Yazeed Sultan Alshahrani, Wejdan Abdullah Alnahdi, Aws Mubarak Algahtany, Naglaa Youssef, Ramy Mohamed Ghazy, Ali Abdullah Alqahtani, Mosab Abdulaziz Deajim

**Affiliations:** 1Faculty of Medicine, King Khalid University, Abha 62529, Saudi Arabia; 441800173@kku.edu.sa (Y.S.A.);; 2Department of Family Medicine, King Khalid University, Abha 62529, Saudi Arabia; 3Faculty of Medicine, Umm Al Qura University, Makkah 21955, Saudi Arabia; 4College of Medicine, Taibah University, Medina 41477, Saudi Arabia; 5Jeddah Eye Hospital, Ministry of Health, Jeddah 11176, Saudi Arabia; 6Department of Medical-Surgical Nursing, College of Nursing, Princess Nourah Bint Abdulrahman University, Riyadh 11671, Saudi Arabia; nfyoussef@pnu.edu.sa; 7Tropical Health Department, High Institute of Public Health, Alexandria University, Alexandria 21561, Egypt; ramy_ghazy@alexu.edu.eg

**Keywords:** breaking bad news, healthcare workers, Saudi Arabia, healthcare worker burnout, communication

## Abstract

Breaking bad news is an intrinsic aspect of physicians’ clinical practices. This study aims to investigate how Saudi physicians manage the process of communicating bad news and explore potential differences in breaking bad news practices between young physicians (interns) and their older colleagues. From 1 March to 15 April 2023, ok an anonymous online cross-sectional survey was conducted to explore the communication practices of Saudi physicians concerning breaking bad news using the Communicating Bad News Questionnaire. The physicians were recruited through convenience and snowball sampling methods, and the survey questionnaire was distributed on various social media platforms, including Facebook, Twitter, LinkedIn, and WhatsApp. Data were analyzed using R version 4.2.1. A total of 782 physicians were included in this study. Male physicians represented 50.9% of the participants. Three-quarters (74.7%) were aged 25–30 years. The largest proportion of physicians (45.3%) were interns, followed by junior residents (22.9%), senior residents (11.0%), and specialists (6.5%). The median years of experience was 1.0, ranging from 0 to 45 years. Regarding the place of work, most physicians (86.6%) worked in hospitals, while 13.4% worked in primary healthcare centers. A total of 14.8% said they were not comfortable with discussing patients’/relatives’ issues (20.60 among interns vs. 10.50% among non-interns, χ^2^ = 27.50, *p* = 0.0001), 66.6% reported being trained to break bad news (59.60% among interns vs. 72.40% among non-interns, χ^2^ = 14.34, *p* = 0.001), 59.1% reported breaking bad news to the patient, 37.9% reported to the family, and 3.1% reported to both, with no significant difference between interns and non-interns. A substantial proportion of physicians reported feeling uncomfortable discussing sensitive issues with patients and their relatives despite having received training to deliver bad news and being willing to communicate bad news directly to patients. Notably, our analysis identified a significant disparity between intern and non-intern physicians, particularly in terms of their comfort level in addressing patient-related concerns and access to breaking bad news training.

## 1. Introduction

In recent years, significant research has been conducted on the role of physician–patient communication, and its importance has been increasingly recognized. Studies have shown that inadequate patient communication can lead to various negative outcomes [1,2]. These include feelings of distress and uncertainty, low levels of satisfaction with medical care, increased levels of anxiety and depression, a sense of helplessness and anger, and a reduction in health-related quality of life (HRQoL) [3,4].

Breaking bad news is an intrinsic aspect of the daily clinical practices of physicians. Breaking bad news is defined as “any information that adversely and negatively affects the patient’s view of the future” [5]. The delivery of this news is of great importance as it can have a significant impact on both patients and physicians, especially when there is a lack of sufficient communication skills and knowledge [6]. Physicians often find that breaking bad news is a complex and stressful situation and can have psychological effects on both the patient and the doctor [7]. Studies have highlighted the need and interest of patients to know the truth [8,9,10]. Consequently, if patients perceive their doctors as dishonest, it can lead to increased anxiety and loss of trust [11]. 

Statistics have revealed a significant communication gap in the realm of cancer care. Research has estimated that 53–64% of the surveyed cancer patients do not receive adequate information about their diagnosis and available treatment options [12,13,14]. This gap is especially pronounced among individuals in advanced stages of illness [14]. The paternalistic doctor–patient relationship and the various opinions and intentions of patients and families regarding disclosure and non-disclosure both contribute to the conflict inherent in the shift toward increased openness toward patients about their diagnosis. [15] Studies have shown that the custom of sharing information with family members that is not shared with patients can lead to arguments between doctors, patients, and relatives [16,17]. For example, Arraras et al. [16] observed that, while 70% of the same population said they would prefer that information about the diagnosis be hidden from a relative who had cancer, 90% of noncancer patients requested a complete or partial awareness of their cancer diagnosis. A study that examined the most common disclosure communication techniques by Gordon and Paci [17] found that less than half of all responding doctors would tell patients about their cancer diagnosis and prognosis if the patient wanted to know, but family members were against the patient’s knowledge.

In fact, being truthful about bad news has several benefits, including strengthening the physician–patient relationship, reducing physician complaints, and improving decision making in the treatment process [18]. However, there are exceptions, such as high-risk situations in which there is a probability of suicide or harm to others. Despite the benefits of truthfulness, if bad news is not delivered properly, it can have negative consequences [19]. Certainly, far from being blamed, unexpected emotional reactions from patients and their families and expressing pity or facing difficult questions are among the reasons that prevent doctors from being truthful about breaking bad news [20].

In the past, the knowledge and practice of revealing bad news had not been well structured and widely taught [5]. However, in recent times, significant progress has been made, and several recommendations have been published to provide clear guidelines for this complex and delicate skill [21,22]. Notably, various protocols, such as the SPIKES protocol published in 2000, have gained popularity and widespread acceptance worldwide, particularly for their application in delivering bad news to cancer patients [23]. These guidelines have contributed to improving the quality of physician–patient communication and improving the way bad news is conveyed to patients and their families. However, published findings highlight discrepancies between recommended approaches and patient expectations [24]. Compliance with the SPIKES protocol was observed at rates of 80% and 84.3% in Korea and Brazil, respectively [25,26]. In Sudan, the majority of doctors usually or sometimes reported adhering to the SPIKES protocol, with a rare instance of non-adherence falling within the range of zero to 13.5% [27]. It is important to note that facing challenges when delivering difficult news is an inevitable outcome. This can often be attributed to the lack of formal training or education, which is a common problem in various countries [28,29]. Importantly, it is worth emphasizing that education alone may not lead to improved communication in the context of breaking bad news; it must be complemented by practical training [30]. These findings underscore the need for further exploration and evaluation of how bad news is handled in clinical practice to ensure that patients’ expectations are met adequately.

In particular, in Saudi Arabia, a large number of cancer patients remain uninformed about the full extent and prognosis of their condition [31]. In 1984, a survey found that only 16% of adult patients were informed about their cancer diagnosis, 34% were told that they had a tumor, and 69% were informed about their cancer diagnosis [32]. A study of 321 physicians and 264 hospital attendees found that 67% preferred to inform the patient over the family about the diagnosis of incurable cancer, and 56% would disclose the information even if the family objected. However, 59% of physicians would inform the family without the patient’s consent for HIV infection diagnoses [33]. Significantly, the vast majority of patients (99%) expressed a strong preference to be fully informed about the details of their disease, with a 100% rejection of the idea of withholding information [33,34]. Indeed, the perspectives of diseases, including cancer, are influenced by various factors, including religious faith, belief systems, societal norms, cultural traditions, and taboos. In Middle Eastern societies, patients often prefer that their families receive information and participate in treatment decisions, aligning with deeply ingrained values of communal support and familial cohesion [35].

The experience of breaking bad news among Saudi Arabian physicians has received little attention. Therefore, it was essential to conduct the current study that aims to investigate how Saudi physicians manage the process of delivering bad news, taking the initial steps towards developing culturally sensitive approaches. Additionally, it aims to explore potential differences in breaking bad news practices between young physicians (interns) and their older colleagues.

## 2. Materials and Methods

### 2.1. Study Design and Setting

An anonymous online cross-sectional survey was conducted from 1 March 2020 to 15 April 2020. Physicians were recruited through convenience and snowball sampling methods, where the questionnaire was distributed across various social media platforms, including Facebook, Twitter, LinkedIn, and WhatsApp. Saudi Arabia comprises 13 emirates: Asir, Al-Bahah, Al-Jawf, Al-Qassim, Eastern, Ha’il, Jazan, Mecca, Medina, Najran, Northern Borders, Riyadh, and Tabuk. From these, a set of three emirates was randomly chosen. Subsequently, two hospitals were selected from each of the three chosen emirates. A focal collaborator was identified in each of these hospital settings. The collaborators were requested to voluntarily distribute the questionnaires. The questionnaire itself was uploaded to a Google form, physicians were recruited using convenience and snowball sampling methods, and the survey questionnaire was distributed on various social media platforms, including Facebook, Twitter, LinkedIn, and WhatsApp. To recruit physicians for this study, a combination of convenience and snowball sampling techniques was employed.

### 2.2. Sample Size and Study Population

In this study, the sample size was determined using G* Power 3.1 with an assumed effect size of 0.1, an alpha error of 5%, and a power of 95%. The minimum sample size required to detect an inappropriate communication prevalence of 50% was calculated to be 321. To account for a potential nonresponse rate of 10% (based on the pilot study), the sample size was increased to 350. Additionally, to stratify respondents into intern and non-intern physicians, we duplicated the sample size to 700. This study included healthcare workers in Saudi Arabia without restricting inclusion based on their subspecialty. To participate in this study, physicians were required to have access to the Internet through smartphones or computers.

### 2.3. Assessment of the Study Outcome

The main outcome of this study was to investigate Saudi physicians’ practices when delivering bad news. Additionally, this research seeks to compare the communication practices between intern physicians and non-intern physicians in such situations.

### 2.4. Data Collection

We used the validated questionnaire for communicating bad news developed by González-Cabrera et al. [36], which was piloted among 15 HCWs to assess their understanding and identify any ambiguities. Valuable feedback from the pilot study was used to improve the clarity and suitability of the questionnaire. Cronbach’s alpha was used to evaluate the internal consistency of the tool, which produced a sufficient coefficient of 0.73. Based on the pilot study, the completion time was estimated to be between 3 and 7 min, and minor modifications were made to ensure clarity. Data from the pilot study were excluded from the main study. Measures were implemented to prevent duplicate submissions, ensuring that each participant contributed to one single response.

The first section of the questionnaire collected sociodemographic data, including gender, age, marital status, job titles, years of experience, and place of work. The second section inquired about physicians’ comfort level in discussing issues related to health status with patients or their relatives, providing response options of “No”, “Not sure”, and “Yes”. Additionally, respondents were asked if they had received training for breaking bad news, with choices of “No”, “Not sure”, and “Yes”. Additionally, physicians were asked about their preferred recipient for delivering bad news, with options for “Both”, “Family”, and “Patient”. The last section included a series of questions related to healthcare care provider communication practices when delivering bad news to patients. The physicians were asked their opinions and actions in various communication scenarios. They were asked whether they believed that patients should be told everything about their condition, whether they chose a quiet and private place beforehand to communicate bad news, and whether they called the patient by their name. Additionally, they were asked about their eye contact with patients during conversations and whether they had inquired about the patient’s prior knowledge before starting the conversation. This section also explores how healthcare providers gauged the patient’s feelings, fears, and worries and how they maintained an attitude of active listening when patients responded with anxiety, fear, sadness, or aggression. The questionnaire covered other aspects, such as showing support and understanding non-verbally, presenting themselves assertively during communication, observing the patient’s emotions following bad news, ensuring that all doubts and questions were addressed, establishing a care plan if needed, and planning for future actions in challenging situations. The responses were recorded on an ordinal scale (always, never, or sometimes).

### 2.5. Ethical Approval

Ethical approval was obtained from King Khalid University in Abha, Saudi Arabia (IRB = HAPO-06-B-001). Informed consent was obtained from all physicians, and they were given the option to either provide consent and participate in this study or decline to participate. Participants received detailed information about this study’s objective, potential emotional impact, and availability of support resources prior to participating in this study. The research team prioritized maintaining the confidentiality and anonymity of personal information from the physicians throughout the research process.

### 2.6. Statistical Analysis

Statistical analysis was performed using the R software version 4.2.1. Categorical variables were presented as counts and percentages. Pearson’s chi-squared test was used to compare two independent categorical variables. The associated degrees of freedom (df) indicate the number of independent categories that contribute to the chi-squared value. The strength of the association between categorical variables was assessed using the Cramér’s V test. Cramér’s V was applied to categorical variables with more than two levels. Cramér’s V test produces measure values ranging from 0 to 1, where 0 indicates no association and 1 signifies perfect association. For this analysis, a *p*-value less than 0.05 was considered statistically significant.

## 3. Results

### 3.1. Sociodemographic Data 

A total of 782 physicians were included in this study. The gender distribution showed that there were 384 female physicians (49.1%) and 398 male physicians (50.9%). In terms of age, three-fourths of physicians (74.7%) were 25–30 years old. Marital status indicated that 68.4% of the physicians were single, 25.1% were married, and the rest were divorced or widowed. Regarding job titles, the largest proportion of physicians (45.3%) were interns, followed by junior residents (22.9%), senior residents (11.0%), and specialists (6.5%). The median years of experience were 1.0, ranging from 0 to 45 years. Regarding the place of work, most physicians (86.6%) worked in hospitals, while 13.4% worked in primary healthcare centers (PHCCs) (Table 1).

### 3.2. Comfort Level in Discussing Bad News with Patients or Their Relatives 

Regarding comfort in discussing patients’/relatives’ issues, most physicians (70.1%) reported feeling comfortable, 14.8% said they were uncomfortable, and 15.1% were unsure. Comparing interns to non-interns, a higher percentage of non-interns (77.8%) compared to interns (60.7%) reported feeling comfortable discussing issues (χ^2^ = 27.503, df = 2, Cramér V = 0.188, *p* = 0.0001) (Figure 1). Most physicians (66.6%) reported being trained, 24.4% were not trained, and 9.0% were unsure. Non-interns (72.4%) had a higher percentage of training than interns (59.6%); this difference was statistically significant. [χ^2^ = 14.342, df = 2, Cramer’s V = 0.135, *p* = 0.001] (Figure 2). The distribution of physicians, when asked about whom they deliver breaking bad news, is shown in Figure 3. The majority (59.1%) reported that they delivered bad news to the patient, 37.9% to the family, and 3.1% reported that they delivered bad news to both. The difference between interns and non-interns in providing bad news to patients or families was not statistically significant [χ^2^ = 3.678, df = 2, Cramer’s V = 0.069, *p* = 0.159] (Figure 3).

### 3.3. The Communication Practices of Healthcare Providers When Break Bad News

Most physicians (78.6%) responded affirmatively that patients should be told everything, while a small proportion (12.9%) disagreed. The remaining physicians (8.4%) were not sure. A significant majority (84.7%) reported selecting a quiet and private place for the conversation, while a smaller percentage (7.2%) did not, and some (8.2%) were uncertain. Most physicians (72.6%) reported calling patients by their names, indicating a personalized approach to communication. However, a notable proportion (24.2%) reported doing so only sometimes. Most (71.7%) reported looking at the patient’s face or eyes during conversations. Few physicians (2.6%) stated they never did, and some (25.7%) reported doing so only sometimes. Most physicians (56.3%) reported always finding out what the patient knows, while some (39.6%) did so sometimes before starting the conversation. A significant percentage (61.6%) reported always establishing a care plan with the patient to address the new situation, while a smaller proportion (33.0%) did so sometimes. There was a significant difference in response between interns and non-interns (χ^2^ = 11.858, *p* = 0.003). A higher percentage of non-interns (77.6%) reported calling patients by their name, compared to 66.7% of interns (Table 2).

## 4. Discussion

Currently, there is a growing focus on teaching the skill of breaking bad news during medical education and the early years of medical practice. Individualized training is considered to be the most effective approach for enhancing communication skills, even at the undergraduate level. Recognizing the significance of this skill, efforts are being made to equip medical students and young healthcare professionals with the necessary knowledge and techniques to effectively deliver bad news to patients and their families [37,38]. This study aimed to assess Saudi physicians’ experiences with and beliefs in breaking bad news. We found that a major section feels uncomfortable discussing patients’/relatives’ issues despite being trained to break bad news to the patient and their families. There is a significant difference between interns and non-interns in feeling comfortable when discussing patients’/relatives’ problems and receiving training to deliver bad news.

Several studies conducted in this area have consistently highlighted the intricate nature of the task, and the majority of clinicians recognize the complexity of this communication skill. Many clinicians also expressed challenges in truthfully conveying difficult information to their patients, underscoring the significance of addressing this skill gap [24,39,40,41]. It is also possible that doctors’ empathy is another factor that hinders patient-doctor communication. Being a recipient of bad news correlates with a lower likelihood of declaring a lack of delivery skills regarding breaking bad news. Empathy for patients can be enabled or prevented by various factors [42]. Notably, Saudi Arabian personality factors explain 10–19% of the variance in empathy [43].

Our findings revealed a notable disparity, particularly among interns who reported a higher level of discomfort when engaging in discussions related to the issues and concerns of patients’ relatives. This difference may be due to the significant difference between interns and non-interns in receiving training related to communicating bad news. This discrepancy underscores the importance of addressing this discomfort through targeted interventions and training programs to enhance the communication skills of medical professionals, particularly during their formative training years. It is worth noting that in many educational environments, a student’s only opportunity to learn is through close observation of role models while they are being trained. Unfortunately, if children have had poor role models, these encounters may not offer them adequate opportunities to learn, but this does not mean that they have acquired the necessary skills [44].

Research has indicated a deficiency in the provision of assistance during the communication of distressing information, as well as insufficiencies in training related to this crucial clinical proficiency [45]. Interestingly, in this study, a substantial portion of physicians reported receiving training to deliver bad news, with a more pronounced trend observed among non-interns. This discrepancy suggests that efforts to provide training in the effective communication of bad news were more prevalent among experienced medical professionals, potentially reflecting a growing recognition of the significance of this skill in medical practice and the need to provide such training to future physicians. Arbabi et al. [46] conducted a study at the Cancer Institute of Tehran University of Medical Sciences, revealing that a mere 8% of physicians had received formal training in revealing bad news. Various methods (active learning, passive learning, mixed approaches, and online learning) are used to instruct medical students, resident physicians, and experienced doctors to deliver bad news. While mixed approaches (practical and theoretical exercises) hold particular significance, it is noteworthy that all strategies yield favorable outcomes [46].

### Strengths and Limitations

This study is of significant importance as it is the first of its kind in Saudi Arabia to explore physicians’ practices to deliver bad news. However, it is essential to acknowledge the limitations of the present study. The use of an online survey could have introduced selection bias, potentially limiting the representation of the broader studied community. Furthermore, the absence of a random sampling method may restrict the generalizability of our findings to larger populations. The design of the cross-sectional survey also limits the ability to establish causality between identified factors and breaking bad news. Despite these limitations, this study provides valuable information on the prevalence and associated factors of bad news among Saudi physicians and serves as a foundational reference for future research and interventions in this field.

## 5. Conclusions

The findings revealed that a substantial proportion of physicians reported feeling uncomfortable discussing sensitive issues with patients and their relatives, having received training in how to deliver bad news, and being willing to communicate bad news directly to patients. Notably, our analysis identified a significant disparity between intern and non-intern physicians, particularly in terms of their comfort level in addressing patient-related concerns and access to breaking bad news training. These insights underscore the need for targeted interventions and training programs, especially for interns, to improve their skills and confidence in effectively managing breaking bad news situations.

## Figures and Tables

**Figure 1 healthcare-11-02528-f001:**
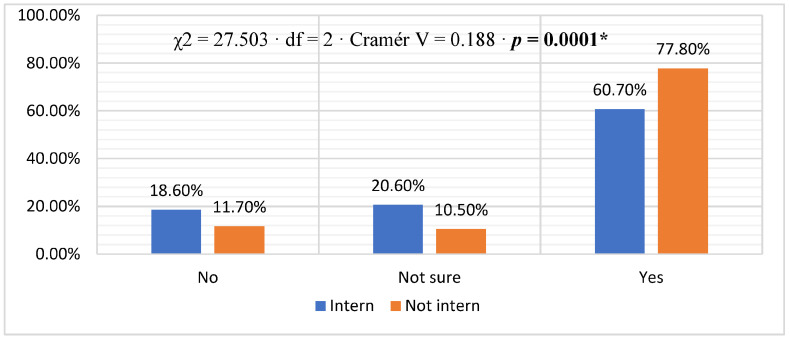
Feel comfortable with discussing patients’/relatives’ issues [* statistically significant *p* < 0.05].

**Figure 2 healthcare-11-02528-f002:**
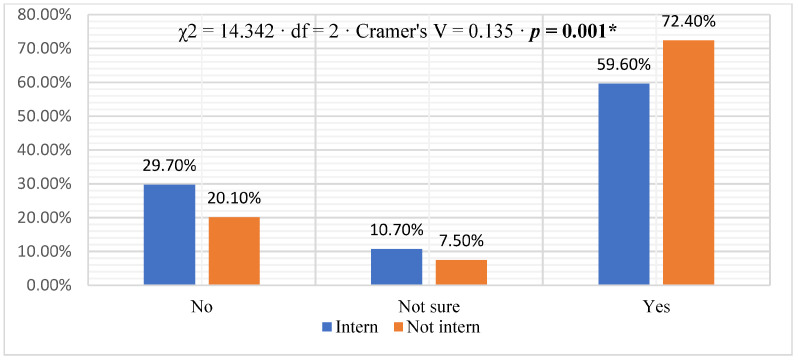
Receiving training on breaking bad news [* statistically significant *p* < 0.05].

**Figure 3 healthcare-11-02528-f003:**
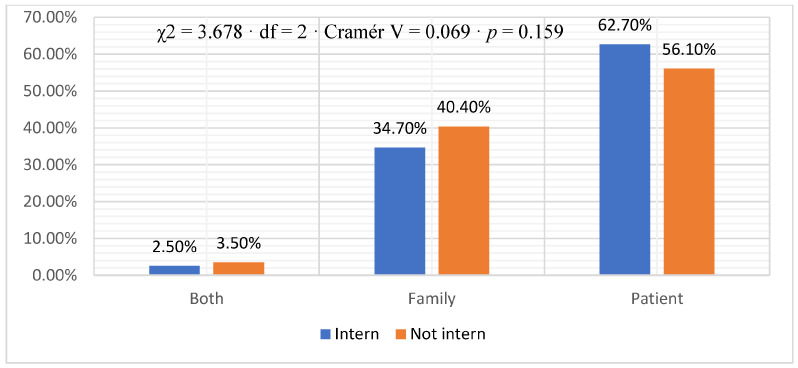
For whom to break bad news.

**Table 1 healthcare-11-02528-t001:** Demographic and professional characteristics of study physicians.

Variables (N = 782)		n (%)
Gender	Female	384 (49.1%)
	Male	398 (50.9%)
Age	25–30 years	584 (74.7%)
	31–35 years	78 (10.0%)
	36–40 years	52 (6.6%)
	41–45 years	31 (4.0%)
	46–50 years	28 (3.6%)
	Above 51 years	9 (1.2%)
Marital status	Divorced	37 (4.7%)
	Married	196 (25.1%)
	Single	535 (68.4%)
	Widow	14 (1.8%)
Job title	Consultant	38 (4.9%)
	Fellow	41 (5.2%)
	Intern	354 (45.3%)
	Junior Resident	179 (22.9%)
	Registrar	33 (4.2%)
	Senior Resident	86 (11.0%)
	Specialist	51 (6.5%)
Years of experience	Median [Min, Max]	1.0 [0, 45]
Place of work	Hospitals	677 (86.6%)
	PHCCs	105 (13.4%)

PHCCs: primary healthcare centers.

**Table 2 healthcare-11-02528-t002:** Communication practices of healthcare professionals when breaking bad news to patients.

Variable		Overall N = 782	Intern n = 354	Non-Intern n = 428	
**Do you feel patients should be told everything about them?**	No	101 (12.9%)	44 (12.4%)	57 (13.3%)	χ^2^ = 3.412, df = 2, Cramér V = 0.066, *p*= 0.182
	Not sure	66 (8.4%)	37 (10.5%)	29 (6.8%)	
	Yes	615 (78.6%)	273 (77.1%)	342 (79.9%)	
**Do you choose a quiet and private place beforehand to communicate bad news?**	No	56 (7.2%)	24 (6.8%)	32 (7.5%)	χ^2^ = 5.631, df = 2, Cramér = 0.085, *p* = 0.060
	Not sure	64 (8.2%)	38 (10.7%)	26 (6.1%)	
	Yes	662 (84.7%)	292 (82.5%)	370 (86.4%)	
**Do you call the patient by their name?**	Always	568 (72.6%)	236 (66.7%)	332 (77.6%)	**χ^2^ = 11.858, df = 2,** Cramér **V = 0.123, *p* = 0.003 ***
	Never	25 (3.2%)	15 (4.2%)	10 (2.3%)	
	Sometimes	189 (24.2%)	103 (29.1%)	86 (20.1%)	
**Do you look at the patient’s face or in the eyes while you talk or listen?**	Always	561 (71.7%)	252 (71.2%)	309 (72.2%)	χ^2^ = 0.109, df = 2 Cramér = 0.012, *p* = 0.947
	Never	20 (2.6%)	9 (2.5%)	11 (2.6%)	
	Sometimes	201 (25.7%)	93 (26.3%)	108 (25.2%)	
**Before starting the conversation, do you find out what the patient already knows about the news that you are going to communicate?**	Always	440 (56.3%)	186 (52.5%)	254 (59.3%)	χ^2^ = 3.716, df = 2, Cramér = 0.069, *p* = 0.156
	Never	32 (4.1%)	15 (4.2%)	17 (4.0%)	
	Sometimes	310 (39.6%)	153 (43.2%)	157 (36.7%)	
**To find out what the patient knows and how much they want to know, do you use questions such as, before I talk, do you want to tell me anything or ask me something?**	Always	444 (56.8%)	190 (53.7%)	254 (59.3%)	χ^2^ = 2.733, df = 2, Cramér = 0.059, *p* = 0.255
	Never	44 (5.6%)	20 (5.6%)	24 (5.6%)	
	Sometimes	294 (37.6%)	144 (40.7%)	150 (35.0%)	
**Before communicating bad news, do you find out in what way the news may affect the patient’s personal, social, or work life?**	Always	439 (56.1%)	186 (52.5%)	253 (59.1%)	χ^2^ = 3.852, df = 2, Cramér = 0.070, *p* = 0.146
	Never	43 (5.5%)	19 (5.4%)	24 (5.6%)	
	Sometimes	300 (38.4%)	149 (42.1%)	151 (35.3%)	
**In the event that the patient is unsure they wish to be informed, do you give the patient time to consider it?**	Always	523 (66.9%)	226 (63.8%)	297 (69.4%)	χ^2^ = 2.717, df = 2, Cramér = 0.059, *p* = 0.257
	Never	25 (3.2%)	12 (3.4%)	13 (3.0%)	
	Sometimes	234 (29.9%)	116 (32.8%)	118 (27.6%)	
**Do you keep in mind the opinion of the patient?**	Always	523 (66.9%)	224 (63.3%)	299 (69.9%)	χ^2^ = 5.488, df = 2, Cramér = 0.084, *p* = 0.064
	Never	35 (4.5%)	14 (4.0%)	21 (4.9%)	
	Sometimes	224 (28.6%)	116 (32.8%)	108 (25.2%)	
**Do you use appropriate language to allow the patient to digest the bad news?**	Always	559 (71.5%)	248 (70.1%)	311 (72.7%)	χ^2^ = 0.660, df = 2, Cramer’s V = 0.029, *p* = 0.719
	Never	33 (4.2%)	16 (4.5%)	17 (4.0%)	
	Sometimes	190 (24.3%)	90 (25.4%)	100 (23.4%)	
**Do you communicate the bad news sequentially and in an organized manner, not giving more information until you are sure that the information already given has been digested?**	Always	515 (65.9%)	222 (62.7%)	293 (68.5%)	χ^2^ = 2.970, df = 2, Cramér = 0.062, *p* = 0.226
	Never	26 (3.3%)	12 (3.4%)	14 (3.3%)	
	Sometimes	241 (30.8%)	120 (33.9%)	121 (28.3%)	
**Do you ask a question to find out how the patient is feeling?**	Always	465 (59.5%)	198 (55.9%)	267 (62.4%)	χ^2^ = 3.741, df = 2, Cramér = 0.069, *p* = 0.154
	Never	53 (6.8%)	24 (6.8%)	29 (6.8%)	
	Sometimes	264 (33.8%)	132 (37.3%)	132 (30.8%)	
**In terms of the feelings, fears, and worries of the patient, do you verbally express your understanding or responsiveness?**	Always	485 (62.0%)	222 (62.7%)	263 (61.4%)	χ^2^ = 2.233, df = 2, Cramér = 0.053, *p* = 0.327
	Never	42 (5.4%)	23 (6.5%)	19 (4.4%)	
	Sometimes	255 (32.6%)	109 (30.8%)	146 (34.1%)	
**When the patient’s response is anxiety, fear, sadness, or aggression, do you maintain an attitude of active listening?**	Always	535 (68.4%)	235 (66.4%)	300 (70.1%)	χ^2^ = 1.595, df = 2, Cramér = 0.045, *p* = 0.451
	Never	28 (3.6%)	12 (3.4%)	16 (3.7%)	
	Sometimes	219 (28.0%)	107 (30.2%)	112 (26.2%)	
**Do you show support and understanding non-verbally?**	Always	468 (59.8%)	198 (55.9%)	270 (63.1%)	χ^2^ = 5.655, df = 2, Cramér = 0.085, *p* = 0.059
	Never	56 (7.2%)	32 (9.0%)	24 (5.6%)	
	Sometimes	258 (33.0%)	124 (35.0%)	134 (31.3%)	
**When you communicate bad news, do you present yourself assertively, expressing your thoughts confidently?**	Always	453 (57.9%)	191 (54.0%)	262 (61.2%)	χ^2^ = 4.194, df = 2, Cramér = 0.073, *p* = 0.123
	Never	38 (4.9%)	19 (5.4%)	19 (4.4%)	
	Sometimes	291 (37.2%)	144 (40.7%)	147 (34.3%)	
**Do you observe the emotions that have emerged in the patient following the communication of bad news?**	Always	496 (63.4%)	219 (61.9%)	277 (64.7%)	χ^2^ = 1.495, df = 2, Cramér = 0.044, *p* = 0.474
	Never	47 (6.0%)	25 (7.1%)	22 (5.1%)	
	Sometimes	239 (30.6%)	110 (31.1%)	129 (30.1%)	
**Do you ensure that at the end of the conversation the patient has no further doubts or questions?**	Always	523 (66.9%)	232 (65.5%)	291 (68.0%)	χ^2^ = 3.363, df = 2, Cramér = 0.066, *p* = 0.186
	Never	28 (3.6%)	9 (2.5%)	19 (4.4%)	
	Sometimes	231 (29.5%)	113 (31.9%)	118 (27.6%)	
**Do you establish, if necessary, a care plan together with the patient to address the new situation?**	Always	482 (61.6%)	209 (59.0%)	273 (63.8%)	χ^2^ = 1.900, df = 2, Cramér = 0.049, *p* = 0.387
	Never	42 (5.4%)	21 (5.9%)	21 (4.9%)	
	Sometimes	258 (33.0%)	124 (35.0%)	134 (31.3%)	
**Do you explore the possible occurrence of challenging situations after the communication of bad news and establish a strategy for future action?**	Always	443 (56.6%)	205 (57.9%)	238 (55.6%)	χ^2^ = 0.681, df = 2, Cramér = 0.030, *p* = 0.712
	Never	53 (6.8%)	25 (7.1%)	28 (6.5%)	
	Sometimes	286 (36.6%)	124 (35.0%)	162 (37.9%)	
**Do you farewell the patient at the end of the conversation?**	Always	531 (67.9%)	228 (64.4%)	303 (70.8%)	χ^2^ = 3.681, df = 2, Cramér = 0.069, *p* = 0.159
	Never	25 (3.2%)	12 (3.4%)	13 (3.0%)	
	Sometimes	226 (28.9%)	114 (32.2%)	112 (26.2%)	

* Statistically significant *p* < 0.05.

## Data Availability

Data are available upon request from the corresponding author via email.

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
