# Peer review of "Examining Saudi Physicians’ Approaches to Communicate Bad News and Bridging Generational Gaps"

_healthcare, 2023, doi:10.3390/healthcare11182528_

Round 1

Reviewer 1 Report

This is an interesting paper, speaking on an important topic around doctors' ability and efficacy in breaking bad news about health to patients and families, and their own views on its impact on their practice and themselves. This paper has good potential, though I feel it does need some revision in terms of framing itself within what is known and why it is necessary, clarifying significance of findings, and engaging more with some other relevant theoretical frameworks around it. I encourage the authors to revise and broaden/strengthen the contribution their paper is making.

Specific comments: 

1. The acronym "BBN" for "breaking bad news" is not meaningful and seems quite lazy and a poor way to save on word count. It results in multiple circumstances where the expression in a sentence makes no sense (example in abstract "59.1% reported delivering BBN to the patient" - delivering breaking bad news? This is poorly expressed.) Suggest rewriting, throughout so that rather than consistently using BBN, using more expressive and contextually-sensible language. 

2. The literature overview at the start of the paper is quite shallow; there is good literature that has been engaged with but it has not been deeply or critically summarised. Most of this section is generalisations with little indicative detail - it is stated that truth is necessary, that breaking bad news is stressful, etc, but there is not enough detail on why or how. On several occasions "studies have shown" is used, but only one study is cited (even though it is a review study). Would expect to see Grassi et al's highly cited work on truth-telling in cancer care here (10.1007/s005209900067). Guidelines and "best practice" have been identified but if, as this paper suggests, they aren't really working for all clinicians, there should be some further critical analysis of why beyond the argument made by one study evaluating one protocol. 

3. When stating "a significant number of cancer patients were still uninformed about the complete extent and prognosis of their conditions" - the study cited here is over 25 years old, and is not a useful indication that this is a current and pressing problem. Are there any more up-do-date references or resources which indicate similar? Also recommend not using "significant" if there are not statistical findings, with demonstrated significance, to back this up.

4. Towards the end of the introduction is the mention of "culturally sensitive approaches" - what does this mean? Does this mean in the context of local ethnic/religious culture in Saudi Arabia? (Given that the argument is made here that more needs to be known about how Saudi physicians break bad news?) If so, it might be good to give a brief overview of the sort of cultural specifics of the physician's social role in Saudi culture, and attitudes towards death/dying/bad news in an ethnic/religious framework. Some references might include doi: 10.1186/s12910-023-00918-9, doi: 10.1097/MPH.0b013e318230dab6. 

5. There is a little lack of clarity in the methods. It states that the survey questionnaire was distributed across various social media platforms, in cluding Facebook, Twitter, LinkedIn, and WhatsApp. Was this before or after the specific selected hospital settings were chosen? Was the recruitment limited to just the employees of those hospitals? 

6. Where was the 10% acceptable non-response rate drawn from? Is this standard/normal for surveys of physicians? If so, support by citing 2-3 other studies from elsewhere which use similar targeted sample survey methodology and have a similar response rate.

7. Good to see ethical approval was sought and granted. Did the team have any other ethical considerations/safety mechanisms in the study, such as information provided within the survey about relevant support services should the participants become distressed in giving their reflections?

8. As I am not a statistician I cannot comment on the validity of the method application/findings. I would suggest more clearly flagging statistically significant findings, however. Also would be good to flag if any other variables came up with statistically-significant relationships, particularly around gender.

9. Why the choice to display average years of experiences as a median - is that meaningful? Could this instead be displayed in ranges (e.g. less than 1, 1-5, 5-10, 10-20, 20-30, 30+) or would perhaps a mode be more meaningful?

10. Discussion should cover any and all associations that came up as statistically significant, and aim to provide supporting information that might suggest why. The discussion here seems quite brief in comparison to the rich detail of the questions asked and trends identified. 

11. One thing that might be useful to reflect on in the discussion is the role/importance of empathy in medical practice; how important is empathy in the ability to break bad news? Is empathy part of the Saudi medical curriculum? Suggested resources: Doi:10.1136/bmjopen-2019-036471; doi:10.1177/0141076818769477.

Mostly the English language expression in this paper is of good quality. Take care with use of BBN acronym as it often makes for sloppy expression around it.

Author Response

Reviewer 1

This is an interesting paper, speaking on an important topic around doctors' ability and efficacy in breaking bad news about health to patients and families, and their own views on its impact on their practice and themselves. This paper has good potential, though I feel it does need some revision in terms of framing itself within what is known and why it is necessary, clarifying significance of findings, and engaging more with some other relevant theoretical frameworks around it. I encourage the authors to revise and broaden/strengthen the contribution their paper is making.

I truly appreciate your thoughtful review of the paper and your valuable insights into its potential for improvement. Your feedback is invaluable in guiding the refinement of the paper to ensure its contribution aligns with the significance of the topic.

Specific comments:

  1. The acronym "BBN" for "breaking bad news" is not meaningful and seems quite lazy and a poor way to save on word count. It results in multiple circumstances where the expression in a sentence makes no sense (example in abstract "59.1% reported delivering BBN to the patient" - delivering breaking bad news? This is poorly expressed.) Suggest rewriting, throughout so that rather than consistently using BBN, using more expressive and contextually-sensible language.

I appreciate your feedback regarding the use of the acronym "BBN" for "breaking bad news" in the paper. Your point about it potentially causing confusion and awkward phrasing is valid, and I understand your concern.

  1. The literature overview at the start of the paper is quite shallow; there is good literature that has been engaged with but it has not been deeply or critically summarized. Most of this section is generalisations with little indicative detail - it is stated that truth is necessary, that breaking bad news is stressful, etc, but there is not enough detail on why or how. On several occasions "studies have shown" is used, but only one study is cited (even though it is a review study). Would expect to see Grassi et al's highly cited work on truth-telling in cancer care here (10.1007/s005209900067). Guidelines and "best practice" have been identified but if, as this paper suggests, they aren't really working for all clinicians, there should be some further critical analysis of why beyond the argument made by one study evaluating one protocol.

Response: thank you for your comment. I appreciate your detailed feedback on the literature overview section of the paper. Your insights highlight areas where the paper can be strengthened and provide a more comprehensive understanding of the topic. All your recommendations were considered.

  1. When stating "a significant number of cancer patients were still uninformed about the complete extent and prognosis of their conditions" - the study cited here is over 25 years old, and is not a useful indication that this is a current and pressing problem. Are there any more up-do-date references or resources which indicate similar? Also recommend not using "significant" if there are not statistical findings, with demonstrated significance, to back this up.

Sorry for inconvenience, corrected

  1. Towards the end of the introduction is the mention of "culturally sensitive approaches" - what does this mean? Does this mean in the context of local ethnic/religious culture in Saudi Arabia? (Given that the argument is made here that more needs to be known about how Saudi physicians break bad news?) If so, it might be good to give a brief overview of the sort of cultural specifics of the physician's social role in Saudi culture, and attitudes towards death/dying/bad news in an ethnic/religious framework. Some references might include doi: 10.1186/s12910-023-00918-9, doi: 10.1097/MPH.0b013e318230dab6.

Response: Yeas religious concern affect patient doctor communication. Based on your recommendation we added the following paragraph. [Indeed, disease perspectives including cancer are influenced by various factors, including religious faith, belief systems, societal norms, cultural traditions, and taboos. In Middle Eastern societies, patients often prefer their families to receive information and participate in treatment decisions, aligning with deeply ingrained values of communal support and familial cohesion.]

  1. There is a little lack of clarity in the methods. It states that the survey questionnaire was distributed across various social media platforms, in cluding Facebook, Twitter, LinkedIn, and WhatsApp. Was this before or after the specific selected hospital settings were chosen? Was the recruitment limited to just the employees of those hospitals?

Sorry for inconvenience. We first selected hospital then focal collaborator to circulate the questionnaire in the selected setting. The responses were restricted to local employee.

  1. Where was the 10% acceptable non-response rate drawn from? Is this standard/normal for surveys of physicians? If so, support by citing 2-3 other studies from elsewhere which use similar targeted sample survey methodology and have a similar response rate.

The determined response rate was based on a pilot study that was conducted before circulation of the questionnaire. It was 90%.

  1. Good to see ethical approval was sought and granted. Did the team have any other ethical considerations/safety mechanisms in the study, such as information provided within the survey about relevant support services should the participants become distressed in giving their reflections?

We have added more details to this section.

  1. As I am not a statistician I cannot comment on the validity of the method application/findings. I would suggest more clearly flagging statistically significant findings, however. Also would be good to flag if any other variables came up with statistically-significant relationships, particularly around gender.

Done

  1. Why the choice to display average years of experiences as a median - is that meaningful? Could this instead be displayed in ranges (e.g. less than 1, 1-5, 5-10, 10-20, 20-30, 30+) or would perhaps a mode be more meaningful?

We used median as data was not normally distributed so we used median and IQR

  1. Discussion should cover any and all associations that came up as statistically significant, and aim to provide supporting information that might suggest why. The discussion here seems quite brief in comparison to the rich detail of the questions asked and trends identified.

Sorry for inconvenience we disused all significant findings.

  1. One thing that might be useful to reflect on in the discussion is the role/importance of empathy in medical practice; how important is empathy in the ability to break bad news? Is empathy part of the Saudi medical curriculum? Suggested resources: Doi:10.1136/bmjopen-2019-036471; doi:10.1177/0141076818769477.
  2. Response: Thank you for this comment. Done

Reviewer 2 Report

The aim of the paper is to investigate how Saudi physicians manage the process of delivering bad news, taking the initial steps towards developing culturally sensitive approaches. Additionally, it also explores potential differences in breaking bad news practices between young physicians (interns) and their older colleagues.

The strength of the study is that it's first of it's kind however given that it is an online survey, it might have selection bias. Additionally, the absence of a random sampling method restricts the generalizability of the findings to a larger population.

Some specific changes:

Line 18: Change "Saudi physicians manage the process of delivering" to "how Saudi physicians manage the process of delivering."

Line 38: Change "communicate bad news directly to patients" to "communicating bad news directly to patients."

Author Response

Reviewer 2

Comments and Suggestions for Authors

The aim of the paper is to investigate how Saudi physicians manage the process of delivering bad news, taking the initial steps towards developing culturally sensitive approaches. Additionally, it also explores potential differences in breaking bad news practices between young physicians (interns) and their older colleagues. The strength of the study is that it's first of it's kind however given that it is an online survey, it might have selection bias. Additionally, the absence of a random sampling method restricts the generalizability of the findings to a larger population.

I acknowledge your observation about the study being the first of its kind in this context. This uniqueness brings with it a valuable contribution to the field, shedding light on a crucial aspect that has not been extensively explored before. Your point regarding the potential selection bias due to the online survey method is valid. Recognizing this limitation is crucial, and it's something I've taken into consideration during the study design. Similarly, the absence of a random sampling method is a limitation that impacts the generalizability of the findings. It's important to acknowledge such limitations to provide a comprehensive understanding of the study's scope. Your feedback underscores the importance of continued improvement and refinement in research methodology to enhance the credibility and applicability of the findings. Thank you for sharing your insights, and if you have any further suggestions or thoughts, I would be keen to hear them.

Some specific changes:

Line 18: Change "Saudi physicians manage the process of delivering" to "how Saudi physicians manage the process of delivering."

Done

Line 38: Change "communicate bad news directly to patients" to "communicating bad news directly to patients."

Done

Reviewer 3 Report

I appreciate your diligent efforts in carrying out this research project. The subject matter strikes me as not only captivating but also highly pertinent to clinical practice. Your chosen research methodology appears suitable, and the substantial sample size you've gathered lends credibility to the outcomes. Your exploration of the study's findings is also commendable.

Nevertheless, I would suggest a review of the journal's formatting guidelines. Ensuring the overall thesis format aligns with the journal's requirements and addressing minor typographical errors should be sufficient for refining your work.

1. Omit subheadings from abstracts.

2. Ensure there is a period at the end of the final sentence in the abstract.

3. Eliminate extra spaces below paragraphs in the body text.

4. Apply an indent to the first line of each sentence.

5. Place citations before a period; for example, change "Line 50 quality of life (HRQoL).[1,2]" to "quality of life (HRQoL) [1,2]."

6. In tables, represent the total number of persons with a capital letter "N" and use a lowercase letter "n" for some of the subjects. For example, change "Variables (n =782)" to "Variables (N =782)" and "N (%)" to "n (%)."

Author Response

Reviewer 1

Comments and Suggestions for Authors

I appreciate your diligent efforts in carrying out this research project. The subject matter strikes me as not only captivating but also highly pertinent to clinical practice. Your chosen research methodology appears suitable, and the substantial sample size you've gathered lends credibility to the outcomes. Your exploration of the study's findings is also commendable.

Nevertheless, I would suggest a review of the journal's formatting guidelines. Ensuring the overall thesis format aligns with the journal's requirements and addressing minor typographical errors should be sufficient for refining your work.

Thank you for critical notice. The typo errors have been corrected and the format has been reviewed according to the journal guideline.

  1. Omit subheadings from abstracts.

The abstract structure has been done according the journal guideline and subheading has been deleted.

  1. Ensure there is a period at the end of the final sentence in the abstract.

Done

  1. Eliminate extra spaces below paragraphs in the body text.

Done

  1. Apply an indent to the first line of each sentence.

Done

  1. Place citations before a period; for example, change "Line 50 quality of life (HRQoL).[1,2]" to "quality of life (HRQoL) [1,2]."

Done

  1. In tables, represent the total number of persons with a capital letter "N" and use a lowercase letter "n" for some of the subjects. For example, change "Variables (n =782)" to "Variables (N =782)" and "N (%)" to "n (%)."

Done

Round 2

Reviewer 1 Report

Thank you to the authors for engaging so thoroughly in revising their manuscript, and for their genuine interest and willingness to consider feedback. 

There are only 2 minor stylistic points I would suggest, but would defer to the editor's recommendation on whether or not they are needed. 

First is when there are statistically significant findings in tables to clearly indicate these (either with **  or in bold) for the reader. 

Second just to thoroughly check that language/expression is correct, the paper reads quite well but a few instances of expression errors (mostly from changing from BBN to articulating the concept of breaking bad news more fully, which is a good change.)

Just to thoroughly check that language/expression is correct, the paper reads well but a few instances of expression errors (mostly from changing from BBN to articulating the concept of breaking bad news more fully, which is a good change.)

Author Response

Thank you Dear Reviwer#1 for your time and effort.

  1. We added Astrix to show the significant findings.
  2. We revised the whole manuscript to improve the language.

Thank you again for your time and effort.